# Extraction of Polyphenols and Valorization of Fibers from Istrian-Grown Pomegranate (*Punica granatum* L.)

**DOI:** 10.3390/foods11182740

**Published:** 2022-09-06

**Authors:** Mihaela Skrt, Alen Albreht, Irena Vovk, Oana Emilia Constantin, Gabriela Râpeanu, Mija Sežun, Ilja Gasan Osojnik Črnivec, Uroš Zalar, Nataša Poklar Ulrih

**Affiliations:** 1Department of Food Science and Technology, Biotechnical Faculty, University of Ljubljana, Jamnikarjeva 101, 1000 Ljubljana, Slovenia; 2Laboratory for Food Chemistry, Department of Analytical Chemistry, National Institute of Chemistry, Hajdrihova 19, 1000 Ljubljana, Slovenia; 3Department of Food Science and Engineering and Applied Biotechnology, Faculty of Food Science and Engineering, University “Dunărea de Jos” of Galați, Domnească Street 111, 800201 Galați, Romania; 4Pulp and Paper Institute, Bogišićeva 8, 1000 Ljubljana, Slovenia; 5Centre of Excellence for Integrated Approaches in Chemistry and Biology of Proteins (CipKeBiP), Jamova 39, 1000 Ljubljana, Slovenia

**Keywords:** pomegranate, pomegranate fruit, pomegranate juice, processing residues, bioactive compounds, antioxidants, phenolics, flavonoids, anthocyanins, fibrous compounds, lignocellulose, papermaking

## Abstract

Pomegranate fruit is an ancient fruit that is used not only because of its deep-red color and tasty arils but also due to the health benefits of its extracts. Pomegranate is a valuable source of bioactive compounds, including colorful anthocyanins and other polyphenols. The main objective of the present study was to gain comprehensive knowledge of the phenolic composition and antioxidative activity of a new pomegranate cultivar, grown in Northwest Istria, a part of the North Adriatic coastal area. Various parts of the pomegranate fruit parts were extracted in 70% ethanol or water. Total phenolic content and antioxidative capacity were respectively determined with Folin–Ciocalteu reagent and ABTS radical. Phenolics were examined and analyzed with TLC, LC-MS, and HPLC. Pomegranate juice was prepared from red arils and after thermal treatment, the stability of anthocyanins was monitored for several months to understand the effect of storage. The highest total phenolics were determined in ethanol pomegranate peel extracts (30.5 ± 0.6 mg GAE/g DM), and water peel extracts exhibited the highest antioxidative activity (128 ± 2 µg TE/g DM). After five months of storage of thermally treated pomegranate juice, 50–60 percentage points increase in anthocyanin degradation was observed. Pomegranate peel was further tested as a sustainable inedible food source for papermaking. Due to the low content of cellulose and the high percentage of extractives, as well as a distinguished texture and appearance, the paper made from pomegranate peel is best suited for the production of specialty papers, making it particularly interesting for bioactives recovery, followed by material restructuring.

## 1. Introduction

Pomegranate (*Punica granatum* L.) is one of the oldest known edible fruit tree species, originating from Iran and the Himalayas in the north of India, and spreading for centuries throughout the rest of the world, already being well established in the Mediterranean in the Age of Antiquity. Mediterranean countries are among the main centers for commercial cultivation of pomegranate which also includes Asian countries (India, Iran), the United States, and in smaller part Argentina as well as several other countries.

The pomegranate plant reproduces sexually via insect-induced self-pollination. In the Mediterranean, it blooms in the period from May to June and the fruits are harvested from the end of August into November. The pomegranate fruit is botanically a true berry (i.e., a fleshy fruit arising from a single flower and bearing numerous seeds), enveloped by a smooth reddish rind (leathery pericarp) with a protruding persistent calix. Internally, the fruit consists of a thick fleshy mesocarp that is divided into a series of chambers separated by thin membranous septa. Each chamber embeds the seeds of the fruit within an individual juicy outer layer (arils) from which the pomegranate juice is produced [1,2].

Pomegranate is an economically important plant, widely consumed in various cultures for thousands of years, where they were particularly valued for their taste and nutritional value, and frequently associated in various religions with fertility [2]. Furthermore, because of the immense potential of health benefits from pomegranate fruit, pomegranate also has a long history as a medicinal fruit [3].

In the past decade, pomegranate juice and peel extracts have been extensively studied for functional components related to human health. Pomegranate juice consumption was demonstrated to directly affect insulin levels in type 2 diabetic patients and could contribute to additional control of glucose levels [4]. Extensive studies were performed to evaluate the anticancer [5,6,7,8,9,10] and antimicrobial activity of pomegranate extracts [11,12,13,14,15,16].

Analysis of bioactive compounds of pomegranate fruits revealed that they are a rich source of phenolic compounds [12,17,18,19]; however, the contents and profiles of phenolic compounds are influenced by the origin source of pomegranate fruits and juices [20,21].

In the past few years, the awareness of health beneficial effects of pomegranate fruit has risen among consumers, which consequently led to higher demand for pomegranate fresh fruit as also pomegranate-based processed products. From 2013 to 2017, total import volume to the EU increased by 28,000 tons (https://unece.org/sustainable-development/press/new-unece-standard-will-boost-international-trade-pomegranate, 12 August 2022). Regarding the high demand for pomegranate fruit, some countries even plan to increase the total area of pomegranate orchards [21].

Although a significant amount of knowledge has been accumulated about phenolic composition and their antioxidative activity from different pomegranate cultivars around the world, there is various availability of data on contained bioactive compounds in the various counterparts of the pomegranate fruit for many cultivars, and we still lack information on the new pomegranate cultivar, grown in the most northern part of the Mediterranean, Northwestern Istria. The main objective of the present study was to gain comprehensive knowledge of the phenolic composition and antioxidative activity of the new pomegranate cultivar grown in the northwest of Istria, including bioactive components contained in various plant parts of the harvested fruits. Different extraction solvents were compared to gain a deeper understanding of the polyphenol, flavonoid, anthocyanin profile and antioxidant capacity, as well as provide a feasible approach for the recovery of specific bioactives. On a further practical side, the duration of the pomegranate juice thermal treatment and long-term storage were studied from the aspect of anthocyanin degradation and color change. Furthermore, to elucidate the opportunities of the residual biomass, remaining after the extraction of bioactive compounds, the pomegranate peel was characterized in terms of structural biopolymers content and fiber properties and demonstrated for inclusion in papermaking recipes.

## 2. Materials and Methods

### 2.1. Chemicals and Reagents

Methanol, acetonitrile (HPLC grade), formic acid, sodium acetate, acetic acid, ethyl acetate, *n*-hexane, sodium carbonate (anhydrous), dichlorophenolindophenol, chloroform, 96% ethanol, 1-propanol, 4-dimethylaminocinnamaldehyde (DMACA), hydrochloric acid (p.a.), sulfuric acid (p.a.), nitric acid (p.a.), and potassium chloride were obtained from Merck (Darmstadt, Germany). Toluen was purchased from Carlo Erba (Chaussée du Vexin, France).

Acetone, trolox (6-hydroxy-2,5,7,8-tetramethyl-chroman-2-carboxylic acid), ascorbic acid, paraffin oil, Folin–Ciocalteu’s phenol reagent, 2,2-azinobis(3-ethylbenzothiazoline-6-sulfonic acid) diammonium salt (ABTS), aluminum chloride hexahydrate, sodium nitrite, sodium chlorite, potassium persulfate, 2-aminoethyl diphenylborinate, boron trifluoride and standards of phenolic compounds: protocatechuic acid, chlorogenic acid, *p*-coumaric acid, *o*-coumaric acid, and (+)-catechin, were purchased from Sigma-Aldrich (St. Louis, MO, USA).

Standards of phenolic compounds, quercetin, gallic acid, ferulic acid, and caffeic acid, were purchased from Merck. Standards of cyanidin-3-*O*-glucoside chloride, pelargonidin-3-*O*-glucoside chloride, pelargonidin-3,5-di-*O*-glucoside chloride, and delphinidin-3-*O*-glucoside chloride were from Extrasynthese (Genay, France). Aqueous solutions were prepared with Mili-Q water (Millipore, Bedford, MA, USA).

### 2.2. Plant Material and Sample Preparation

Pomegranate fruits (*Punica granatum* L.) were collected at maturity from a pomegranate plantation at Marasa (45°09′31.1″ N 13°42′27.1″ E, Istria, Croatia). The fruits were washed in water and dried and then manually peeled (separating the peel (pericarp and calyx) from the mesocarp). The flashy arils were carefully separated from the seeds and the arils were further manually juiced. The peel, seeds, and juice were lyophilized and stored at −20 °C until further use.

### 2.3. Preparation of the Pomegranate Water and Ethanol Extracts

Water and ethanol extracts were prepared from the lyophilized samples. For the peels, mesocarp, and seeds, the lyophilized material was crushed (A 11 analytical mill, IKA, Staufen im Breisgau, Germany) to a fine powder prior to extraction, whereas the lyophilized juice was extracted without crushing.

For the two-step extraction, the solvent (4 mL water or 70% ethanol per g lyophilized powder) was added to the samples and the suspensions were shaken for two hours at room temperature. Then, the suspensions were centrifuged at 12,000× *g* for 10 min and the supernatants were decanted and replaced with the equivalent amount of fresh solvent. The same extraction procedure was repeated one more time, supernatants from the first and second fractions were combined and completely dried in a rotary evaporator until further analysis. Prior to subsequent spectrometric or HPLC analysis, dry extracts were diluted in an appropriate solvent.

### 2.4. Total Phenolics

Total phenolics were determined according to the modified method by Gutfinger [22]. Briefly, the Folin–Ciocalteu reagent was first diluted with water (1:2, *v*/*v*) and 125 µL of the diluted reagent was mixed with 200 µL of either water or 70% ethanol extract. After 3 min, 125 µL of 20% (*w*/*w*) Na_2_CO_3_ and water was added to the final volume of 1 mL. After a further 40 min at room temperature, the reaction mixture was centrifuged for 10 min at 8500× *g*. The absorbance of the supernatant was measured at 765 nm against a blank containing water or 70% ethanol. Total phenolics were expressed as mg gallic acid equivalents (GAE)/g of the lyophilized crude extract (g DM).

### 2.5. Total Flavonoids

The total flavonoid content was determined as described by Yang et al. [23]. Briefly, the water or ethanol extracts were diluted with water (1:5, *v*/*v*) and subsequently mixed with 75 µL of 5% (*w*/*w*) sodium nitrite. After 5 min, 150 µL of 10% (*w*/*w*) aluminum chloride was added. After a further 6 min, 500 µL of 1 M sodium hydroxide and water was added to the final volume of 3 mL. The absorbance of the mixture was measured immediately at 510 nm wavelength against a prepared blank. The flavonoid content was determined by a catechin standard curve and expressed as the mean of milligrams of catechin equivalents (CE)/g of the dry lyophilized crude extract.

### 2.6. Antioxidant Capacity

The antioxidant capacity was determined with an optimized method with ABTS, as reported previously by Re et al. [24]. The ABTS (7 mM) was prepared as a stock solution with 2.45 mM potassium persulfate (final concentration), which was left to react for 12 h at room temperature in the dark. Before use, this ABTS stock solution was diluted with distilled water to an absorbance of 0.70 ± 0.02 units at 734 nm. Then, 10 μL of appropriately diluted pomegranate extract was added to 1 mL of diluted ABTS, and the samples were incubated at room temperature for 30 min. The absorbance was measured at 734 nm against a water blank. Trolox was used as an antioxidant standard. The antioxidant capacity was calculated as Trolox equivalent antioxidant capacity (TEAC) and expressed as μg Trolox equivalent (TE)/g of the lyophilized crude extract.

### 2.7. Quantitative Determination of Catechin with TLC

(+)-Catechin was determined in pomegranate extracts with the method developed by Vovk et al. [25]. HPTLC 20 cm × 10 cm cellulose plates (Merck, Art. No. 1.05786) were predeveloped with water. Different volumes of ten times diluted pomegranate extracts prepared by 70% ethanol (1–6 µL) and (+)-catechin standard (2–30 ng on the plate) were applied as 6 mm bands, 10 mm from the bottom of the plates with an Automatic TLC Sampler 4 (Camag, Muttenz, Switzerland). HPTLC plates were developed to a distance of 7 cm in a horizontal developing chamber (Camag) using a sandwich configuration and 6 mL propanol–water–acetic acid (4:2:1, *v*/*v*/*v*) as the developing solvent. The developed plates were dried in a stream of warm air for 2 min and then immersed for 2 s into DMACA dipping detection reagent using Camag immersion device III.

The detection reagent was prepared by dissolving 60 mg of DMACA in 160 mL of cold ethanol. After the addition of 13 mL of concentrated hydrochloric acid, the volume was made up to 200 mL with ethanol [26]. Drying in a stream of warm air for 2 min furnished colored bands for separated compounds.

Documentation of chromatograms was performed 10 min after derivatization by Camag Digistore 2 Documentation system. Densitograms were scanned with a TLC Scanner 3 (Camag) set in the absorption/reflectance mode at 655 nm; the slit length was 4 mm and the width 0.45 mm, and the scanning speed 20 mm s^−1^. Both instruments were controlled by the winCATS program (Version 1.4.1.8154). Catechin content was determined by the catechin calibration curve, y = −0.1015·x^2^ + 9.725·x − 6.65, where y represented peak height and x, catechin mass in ng.

### 2.8. Hydrolysis of Pomegranate Extracts for TLC Screening of Phenolic Acids

The acid hydrolysis of tested pomegranate extracts was performed with a modified method described by Nuutila et al. [27]. For the initial optimization of the procedure, 50 mg of the pomegranate extracts were mixed with 5 mL of a mixture of hydrochloric acid–water–methanol (1:4:5; *v*/*v*/*v*) and 2 mg of ascorbic acid and incubated for 24 h at 80 °C in the Carousel 12 Plus Reaction Station™ (Heidolph North America, Wood Dale, IL, USA). The hydrolysis process was monitored by TLC analyses of the samples taken after 2, 4, 6, 8, 10, 12, 18, and 24 h. Analyses were performed as described in Section 2.9.

### 2.9. TLC Screening of Phenolic Acids

TLC screening of phenolic acids was performed using the method of Simonovska et al. [28]. HPTLC 20 cm × 10 cm silica gel 60 plates (Merck, Art. No. 1.05641) were predeveloped in chloroform–methanol (1:1, *v*/*v*) and dried at 110 °C for 30 min. Solutions of ten times diluted pomegranate extracts prepared by 70% ethanol, solutions of hydrolyzed pomegranate extracts (Section 2.8) and standards (0.1 mg/mL) of phenolic acids (protocatechuic acid, chlorogenic acid, *p*-coumaric acid, *o*-coumaric acid, gallic acid, ferulic acid and caffeic acid), and quercetin were applied as 5 mm bands, 10 mm from the bottom of the plates using an Automatic TLC Sampler 4 (Camag).

Plates were developed by 6 mL *n*-hexane–ethyl acetate–formic acid (20:19:1, *v*/*v*/*v*) in a horizontal developing chamber (Camag) using a sandwich configuration, as well as with ethyl acetate–water–formic acid (85:15:10; *v*/*v*/*v*) in a horizontal developing chamber using tank configuration (15 mL of developing solvent in a tank). The developing distance was 6 cm. The developed plates were dried in a stream of warm air for 2 min and then immersed for 5 s into a 1% methanol solution of diphenylboric acid 2-aminoethyl ester (natural product reagent). The detection at 366 nm and documentation of chromatograms by Camag Digistore 2 Documentation was done after enhancement and fixation of fluorescence by dipping the plates into paraffin–*n*-hexane (1:2, *v*/*v*).

### 2.10. Solid-Phase Extraction of Pomegranate Extracts before LC-MS Analysis and HPLC

Individual anthocyanins were analyzed according to the optimized method of Lätti et al. [29]. The extracts were cleaned up as follows: the samples of the pomegranate extracts were dried using a rotavapor (Buchi R-210), and then resuspended in 1 mL of 3% formic acid. The samples were then loaded onto 1 g Sep-Pak C18 cartridges (Waters, Milford, MA, USA), previously activated with 3 mL of pure methanol and 5 mL of 3% formic acid. After the cartridges had been washed with 6 mL of 3% formic acid, the anthocyanins were eluted with 5 mL of pure methanol [30]. The eluates were evaporated to dryness and redissolved in 0.5 mL of 3% formic acid.

### 2.11. Liquid Chromatography-Mass Spectrometry

Anthocyanins were identified with the HPLC-PDA-MS method by using Accela 1250 UHPLC system coupled to LTQ Velos system ((+)-HESI) (both ThermoFisher Scientific, Waltham, MA, USA), as previously reported [31]. The MS settings used were: T_cap_, 350 °C; T_source_, 325 °C; sheath gas, 60 a.e.; auxiliary gas, 10 a.e.; sweep gas, 3 a.e.; source voltage, 3 kV; and scan range, *m/z* 260–800. The data-dependent tandem MS (MS/MS) analyses were performed on the 1st, 2nd, and 3rd most-abundant ions in the MS spectrum, respectively, using the following settings: isolation width, 2 m/z; normalized collision energy, 35%; and activation time, 10 ms.

The samples were prepared in 30% methanol. The mobile phase consisted of solvent A (5% formic acid) and solvent B (methanol: acetonitrile, 80:20, *v*/*v*). The gradient used was: 0–15 min, 5% to 30% B; 15–17 min, 30% to 50% B; 17–18 min, 50% to 70% B; 18–20 min, 70% B; 20–21 min, 70% to 5% B; 21–25 min, 5% B. The separations were carried out on a Gemini C6-Phenyl 3 µm column (150 mm × 4.6 mm ID) (Phenomenex, Torrance, CA, USA). The HPLC settings were as follows: flow rate 0.6 mL/min; T 45 °C; injection volume 5 µL, with absorbance measured at 520 nm.

### 2.12. Quantification of Individual Anthocyanins

Individual anthocyanins cyanidin-3-*O*-glucoside (Cy-3-Gly), cyanidin-3,5-di-*O*-glucoside (Cy-3,5-diGly), pelargonidin-3-*O*-glucoside (Pel-3-Gly), pelargonidin-3,5-di-*O*-glucoside (Pel-3,5-diGly), delphinidin-3-*O*-glucoside (Del-3-Gly), and delphinium-3,5-di-*O*-glucoside (Del-3,5-diGly) were determined by HPLC 1260 Infinity system (Agilent Technologies, Palo Alto, CA, USA), comprising of a 1260 binary pump (G1312B), 1260 HiPALS autosampler (G1367E), 1260 DAD detector (G4212B) and HPLC 2D ChemStation SW (revision B.04.03). Separations were carried out with a Zorbax Eclipse Plus C18 column (4.6 × 150.0 mm, 3.5 μm, Agilent), protected by Eclipse XDB–C18 security guard cartridge column (4.6 × 12.5 mm, 5 µm, Agilent). The mobile phase consisted of 3% (*v*/*v*) formic acid (A) and of acetonitrile-methanol (85:15, *v*/*v*) mixture (B). The anthocyanins were separated with the following gradient; 0–13 min, 5–8% B; 13–25 min, 8–9% B; 25–45 min, 9–13% B; 45–46 min, 13–100% B; 46–48 min, 100% B; 48–49 min, 100–5% B; 49–55 min, 5% B. The HPLC settings were as follows: flow rate 0.8 mL/min; T 40 °C; injection volume was 40 µL, with absorbance measured at 520 nm (A520). Each analysis was carried out in triplicate. The anthocyanins were identified based on their retention time. The quantification of the individual anthocyanin was determined by external standards in the concentration range from 0.5 to 200 µg/mL. For each standard, the 5-point calibration curve was prepared in the same matrix as pomegranate extracts, and each solution for the 5-point calibration curve was analyzed in triplicate. For each anthocyanin standard, the linear regression correlation factor, R^2^, was determined as well as the limit of detection, LOD, and limit of quantification, LOQ (Table 1).

### 2.13. Thermal Treatment of Pomegranate Juice

Fresh, non-clarified pomegranate juice was pipetted into 1 mL sterile centrifuge tubes, closed, and heated in a Dry Bath System (Star Lab, Hamburg, Germany) for 5, 10, and 20 min at 80 °C. After thermal treatment, samples were put on ice and centrifuged for 5 min at 15,000× *g*. The clear supernatants were filtered through a PTFE filter (0.45 µm) and analyzed with HPLC-DAD as described in Section 2.12. The same thermal treatment and analysis were applied for (i) an unheated control of fresh, non-clarified pomegranate juice, (ii) as well as a mixture of individual anthocyanin standards that was adjusted to the pH of the pomegranate juice.

On a bigger scale, thermal treatment of 0.5 L of fresh, non-clarified pomegranate juice was performed for 10 min at 80 °C, with induction heating (Electrolux, Stockholm, Sweden). The treated juice was stored in sterile 50 mL centrifuge tubes, either overnight at 6 °C or for 3 to 5 months at 18 °C, which were followed up by centrifugation and HPLC analysis.

### 2.14. Color Measurement

Changes in the color were measured at 25 °C with an instrumental colorimeter (Chroma meter CR-400; Konica Minolta, Japan). The parameters measured identified the differences in lightness and darkness (parameter L*), red and green (parameter a*), and yellow and blue (parameter b*). The differences in hue (ΔE) and color intensities (ΔC) were calculated as cumulative values. For further description of the parameters and calculations, see our previous publication [32].

### 2.15. Chemical Composition

In biomass samples, cellulose, hemicellulose, and lignin content were determined with standard or established isolation procedures, filtered through a medium density glass crucible, dried to constant weight at 105 °C, weighed, and expressed as % of dry mass of the samples.

Cellulose content was determined by the Kürschner–Hoffer method. For this, 1 g of the sample residue, which has previously undergone extraction with ethanol, was mixed with 25 mL of the nitration mixture (20 mL of 65% nitric acid and 80 mL of ethanol) and boiled under reflux for 1 h. The process was repeated three times, then the mixture was removed and 100 mL of distilled water for 30 min. The filtered insoluble residue was washed with ethanol and hot water and dried.

Hemicellulose was determined according to the TAPPI 149-75 method (chlorite method) [33]. Here, 0.5 g of the previously extracted sample residue was added to 60 mL of water, 100 μL of glacial acetic acid, and 0.5 g of sodium chlorite and shaken for 1 h at 70 °C. Then, the same amounts of glacial acetic acid and sodium chlorite were added and reheated under the same conditions with occasional stirring. After three consecutive replicates, the mixture was cooled and the solid residue (holocellulose, i.e., cellulose + hemicellulose) was filtered out and dried.

Lignin was determined as Klason’s lignin (acid-insoluble lignin) after pre-extraction with ethanol, according to TAPPI T222 [34]. To 1 g of sample, 15 mL of 72% sulfuric acid was added at room temperature to hydrolyze cellulose and hemicellulose to simple sugars. After 2 h water was added to the sample to reduce the acid concentration to 3%. The mixture was allowed to boil for 4 h, filtered, washed with hot water, and dried at 105 °C.

### 2.16. Delignification, Fiber Characterization, and Paper Production

Fibers were isolated from the rough biomass in the delignification process (removal of the lignin that is binding the fibers together), carried out in a 5 L rotation autoclave digester (Universal Engineering Corporation, Uttar Pradesh, India). Delignification of pomegranate peels was carried out for 3 h at 160 °C, with the addition of 18% NaOH and 6% Na_2_S) [35]. This was followed by Sommerville-type screening (Universal Engineering Corporation, India) with 45 mm × 0.15 mm slots.

Delignified biomass was further disintegrated in a laboratory disintegrator when required, and then screened by the Sommerville (Universal Engineering Corporation) laboratory fractionator using a sieve plate with 45 mm × 0.15 mm slots.

Obtained fibers were further characterized for their morphological and technological papermaking properties (grammage (ISO 536), thickness (ISO 534), tensile properties (ISO 1924–2), tearing resistance (ISO 1974), bursting strength (ISO 2758), roughness according to Bendtsen (ISO 8791–2), ISO whiteness (ISO 2470–1), and opacity (ISO 2471)).

Morphological properties such as fiber length, width, and cell wall thickness were also determined in fiber suspension with a fiber analyzer (Valmet Automation Inc., Espoo, Finland).

Finally, to better understand the properties of the fibers from the pomegranate peel, the obtained delignified material was tested for paper production. Test paper sheets were prepared on a Rapid-Köthen sheet forming machine (Paper Testing Instruments, Laakirchen, Austria). The obtained optical and mechanical properties of paper made from pomegranate peel were compared to typical commercial cellulose for papermaking (Conifers SA Pöls (Zellstoff Pöls AG, Austria) and Eucalyptus Navia (Spain).

### 2.17. Statistical Analysis

Student’s *t*-tests were performed to differentiate between the means, with 95% confidence interval (significance, *p* < 0.05, as indicated). The degradation curves were standardized to the degradation rate of the reference curve (as indicated) and compared to the reference curve using the similarity factor (f2) according to the guidelines of the US FDA guidelines (FDA/CDER, 1997). Calculations were performed using the OriginPro 2018 SR1 b9.5.1.195 software package (OriginLab), as described in our previous publication [36]. For anthocyanins determined by the HPLC method, the linear regression and correlation factor R^2^ were determined in Excel 2017 with the Data Analysis tool and ANOVA. LOD and LOQ were calculated according to the International Council for Harmonisation of Technical Requirements for Pharmaceuticals for Human Use (ICH) guidance on the validation of analytical procedures. LOD was expressed as 3.3xσ/S and LOQ as 10xσ/S, where σ is the standard deviation of response and S is the slope of the calibration curve.

## 3. Results and Discussion

During fractionation of the pomegranate fruit, the obtained fresh fractions were determined. In a sample of 10 fruits with a total fresh mass of 2330 g, individual fruits varied from 123 to 347 g. Upon separation, 1057 g of arils represented half and 640 mL of juice contributed to a third of the initial fruit mass (Table 2).

During further extraction, approximately 4 g of all fresh samples was mixed with 25 mL of the extraction solvent. A lower amount of extractables were extracted with water (20–80%) than ethanol (30–82%) and apart from juice, which is an extract in itself, the mesocarp provided the highest amounts of dry extractables (65–75%).

### 3.1. Total Phenolic Compounds, Flavonoids, and Antioxidant Capacity of Pomegranate Extracts

Profiles of bioactive compounds were markedly different in water or ethanol extracts, as well as different pomegranate fruit fractions. As shown in Table 3, the content of total phenolic compounds for water extracts of the peel, mesocarp, seeds, or juice ranged from 0.4 to 8.8 mg GAE/g dry extract and increased from 1.5 to 30.5 mg GAE/g dry Extract for ethanol extracts. The peel had the highest total phenolic content, with 8.8 or 30.5 mg GAE/g dry extracts, respectively obtained in water or ethanol.

The peel and monocarp exhibited similarly high contents of total phenolic compounds and flavonoids, instead of low bioactives content in seeds and juice. In ethanol, higher amounts of phenolic compounds and flavonoids were extracted, which can be expected from the higher solubility of phenolic compounds in organic solvents. Where the overall phenolic content was low (juice and seeds), ethanol and water extracts contained similar concentrations.

Other studies that have examined bioactive compounds in pomegranate have reported variation in total phenolic content based on geographic origin, analyzed fraction, and type of extraction, where more technologically or chemically intensive extractions procedures generally led to higher recovery of polyphenols (Table 4). The most pronounced differences in total phenolic content could be attributed to genotype, growing conditions, and the choice of extraction solvent and conditions [21,37,38,39].

Furthermore, comparing our extracts in terms of flavonoid content, ethanol pomegranate extracts presented high concentrations of total flavonoids ranging from 0.100 mg CE/g dry juice extract to 4.25 mg CE/g dry peel extract (Table 3). The total flavonoid content was more than doubled in ethanol pomegranate extracts of the peel, mesocarp, and juice and seven times higher in ethanol pomegranate extracts of seeds compared with total flavonoids content in water pomegranate extracts. The flavonoid–phenolic ratio from Table 3 suggests that there are proportionately more flavonoids among phenolic compounds in the water extract of the peel in comparison to its ethanol extract.

Much higher concentrations of total flavonoids were reported for extracts prepared from the peel of 9 pomegranate cultivars with Soxhlet extractions [40]. Namely, these peel extracts ranged from 18.61 to 36.40 mg CE/g extract in polyphenol content. However, similar flavonoid/polyphenol ratios (on average, 0.16) were reported by the same study that was determined in our ethanol extracts, indicating a similar relationship where the extraction strongly relies on organic solvents. In the study of Orak et al. [41], among the studied pomegranate variety Hicaznar and three other genotypes, developed from the gene pool of Hicaznar, Cekirdeksiz, Fellahyemez, and Ernar varieties, the total phenolic content in peel ethanol extracts varied from 132 to 160 mg GAE/g extract and total flavonoids were 3-fold to almost 4-fold higher than found in our study. These results suggested that pomegranate varieties from Antalya are richer in phenolic compounds, although these differences could be also due to the ethanol used as a solvent for the extraction.

The peel and mesocarp extracts also exhibited the highest antioxidant capacity (Table 3). Although higher concentrations of phenolics and flavonoids were extracted by ethanol, the dry water extracts exhibited higher antioxidant capacities. Antioxidative capacity of pomegranate extracts exhibited TEAC values that were the highest in the order of the ethanol peel extract > ethanol mesocarp extract > water peel extract > water mesocarp extract > water juice extract > ethanol seed extract > ethanol juice extract > waster seed extract.

Comparing the higher antioxidant capacity of water extracts (i.e., in average ~10 µg TE/mg GAE) vs. the lower antioxidant capacity of ethanol extracts (i.e., in average ~1.5 µg TE/mg GAE) indicates that chemically different polyphenols were extracted in different solvents, and/or that different additional antioxidants were present. This discrepancy between the analyzed polyphenol/flavonoid content and the antioxidant capacity was particularly evident in the pomegranate juice (being a water extract in its basic form), where a disproportionally high TEAC antioxidant capacity was determined.

Other studies reported higher TEAC values for mesocarp methanol extracts in comparison to the peel methanol extracts and the lowest TEAC values for pomegranate juices [19]. In another assay system, Gil et al. [42] also showed different TEAC values for different groups of phenolic compounds in pomegranate juice. The TEAC value decreased in the order from punicalagins > anthocyanins > hydrolyzable tannins > ellagic acids. In the present study, we can further conclude from Table 3 and Table 7 that TEAC values correlate with flavonoid content, more specifically with anthocyanins, rather than total phenolic content.

### 3.2. Quantitative Determination of Catechin and Qualitative Identification of Phenolic Compounds with TLC

Qualitative and quantitative analyses of flavan-3-ol catechin in 70% ethanol pomegranate extracts were performed on cellulose stationary phase by HPTLC-densitometric method, which included post-chromatographic derivatization with DMACA detection reagent (specific for flavan-3-ols and proanthocyanidins [26]). Catechin was detected in all extracts (see Appendix A). The highest content of catechin was determined in pomegranate peel extract (0.43 mg/g DM extract), followed by seeds ≈ mesocarp extracts and the lowest amount of catechin was determined in juice (0.024 mg/g DM extract) (Table 5).

Five additional phenolic compounds were detected (Table 5) by HPTLC analyses on silica gel stationary phase. Quercetin was detected in all pomegranate extracts, but no phenolic acids were detected in non-hydrolyzed extracts. Upon further treatment of the samples, chlorogenic acid, gallic acid, *o*-coumaric acid, and caffeic acid were also detected after 18 h of hydrolysis. These results conclude that the analyzed phenolic acids are not present in our pomegranate extracts as free acids. The same phenolic acids with additional ferulic acid were detected after hydrolysis at 85 °C in 1.2 M HCl with HPLC-DAD by Karakaplan and Özcan [43].

### 3.3. LC-MS Analysis of Pomegranate Extracts

In this part of the study, seven anthocyanins were detected in pomegranate peel, mesocarp, and juice extracts. Table 6 lists the identified anthocyanins, chromatographic retention times, and MS^2^ fragmentation pattern for further confirmation of their identity. The anthocyanin profiles were identical for the water and ethanol pomegranate extracts but slightly different for the peel and juice extracts. Cyanidin-3-glucoside and pelargonidin-3-glucoside were identified in pomegranate peel, mesocarp, and juice, while delphinidin-3-glucoside was identified only in peel and juice. All other anthocyanins were tentatively assigned using MS and data-dependent MS/MS spectra.

The fragmentation patterns in most cases indicated the loss of a specific moiety (e.g., one hexose [M-162]^+^, two hexoses [M-162-162]^+^, and the core aglycon. Dihexoside derivates of cyanidin (peak 1 in peel and peak 2 in juice), pelargonidin (peak 3 in peel and peak 4 in juice) and delphinidin (peak 1 in juice) were also detected. According to the previous research and their similar anthocyanin elution profiles [19], we could assume that the first two eluted anthocyanins and most polar anthocyanins are 3,5-diglucoside derivates of delphinidin and cyanidin, respectively. In our study, the delphinidin 3-*O*-glucoside eluted before pelargonidin-3,5-diglucoside.

This elution profile was also established with HPLC analysis of individual anthocyanins in our study. Fischer et al. [19] and Sentandreu et al. [18] reported different elution profiles for these two anthocyanins from pomegranate juice. As least polar anthocyanin, pentoside mono-glycoside derivate of cyanidin was also detected in pomegranate peel extract (peak 6) and pomegranate juice extract (peak 7) (see Appendix A). This component was tentatively identified as cyanidin arabinoside, most probably as cyanidin-3-arabinoside, or cyanidin xyloside, also most probably as cyanidin-3-xyloside. Cyanidin-pentoside was also detected by Fischer et al. [19] in pomegranate juice of Peruvian pomegranate fruits of an unknown cultivar.

Sentandreu et al. [18] also reported cyanidin 3-pentoside in pomegranate juice of pomegranate ‘Wonderful’ cultivar. In this study, six major peaks correspond to the well-known 3-glucoside and 3,5-diglucoside derivates of the anthocyanidins delphinidin, cyanidin and pelargonidin and a less known cyanidin-pentoside were described. Although 65 anthocyanins were reported, in our study, more complex anthocyanins derivates were not detected.

### 3.4. HPLC Determination of Individual Anthocyanins in Pomegranate Extracts

Pomegranate anthocyanins in water and ethanol peel and juice extracts were determined by comparing their retention times and UV–Vis spectra with reference compounds. Table 7 shows individual anthocyanins that were quantified and Appendix A shows a typical HPLC chromatogram of pomegranate juice anthocyanins recorded at 520 nm. In this study, we quantitatively determined delphinidin-3-*O*-glucoside (peak 3), pelargonidin-3,5-*O*-di-glucoside (peak 4), cyanidin-3-*O*-glucoside (peak 5), and pelargonidin-3-*O*-glucoside (peak 6). Peaks 1 and 2 were assigned as diglycosylated derivates of delphinidin and cyanidin, respectively, and peak 7 was assigned as a pentoside derivate of cyanidin according to the LC-MS analysis.

Cyanidin-3-*O*-glucoside was found to be the main anthocyanin in both juice and peel water and ethanol extracts, followed by cyanidin-3,5-*O*-di-glucoside. Significant individual anthocyanins in pomegranate peel extracts are cyanidin-3-glucoside, cyanidin-3,5-*O*-diglucoside, and pelargonidin-3-glucoside. Significant individual anthocyanins in pomegranate juice extracts are cyanidin-3,5-*O*-di-glucoside, cyanidin-3-glucoside, and delphinidin-3,5-*O*-di-glucoside.

Anthocyanins maintained similarly low levels in water and ethanol juice extract; however, no diglycosylated derivate of delphinidin was detected in peel extracts and all the other anthocyanin content was considerably higher in peel extracts as compared to juice. Furthermore, more than ten times higher concentrations of individual anthocyanins were measured in water peel extracts than in ethanol peel extracts.

The respective total six anthocyanins content in water and ethanol peel extracts amounted to 295 and 45 µg/g when the content in the extracts was calculated per the total dry mass of the peel. Fischer et al. [17] reported 447 µg of a total of ten anthocyanins/g dry mass of peel. The differences could be due to the different extraction procedures: Fischer et al. [17] used aqueous methanol for the extraction and in our study water or 70% ethanol were used and also due to the different pomegranate varieties used in both studies. Cyanidin-3-*O*-glucoside and pelargonidin-3-*O*-glucoside that were previously identified in pomegranate mesocarp extracts with LC-MS were not detected by HPLC. As many studies include the mesocarp as a part of the peel fraction (Table 2 and Table 3), it is unclear if these more diverse anthocyanin profiles are due to regional/cultivar plant characteristics, analytical approach, or due to fractionation of the fruit parts.

### 3.5. Thermal Treatment of Pomegranate Juice

The experimental conditions for thermal treatment of pomegranate juice in this study may be regarded as conventional thermal treatment in pomegranate processing. In particular, maintaining the pomegranate juice for 5, 10, and 20 min at 80 °C corresponds to the conditions of high temperature long-time (HTLT) pasteurization. Under these practically relevant conditions, we then examined the evolution of four anthocyanins identified as major anthocyanin compounds (delphinidin-3,5-*O*-di-glucoside, cyanidin-3,5-*O*-di-glucoside, delphinidin-3-*O*-glycoside, and cyanidin-3-*O*-glucoside) in pomegranate juice from the sampled unknown Istrian cultivar.

Anthocyanin content that was determined after thermal treatment was compared to the contents of the remaining anthocyanins determined in a mixture of anthocyanin standards exposed to the same conditions (Figure 1).

Degradation of all analyzed anthocyanins increased with the duration of thermal treatment time was the highest at 20 min. The highest degradation in pomegranate juice was observed for delphinidin-3,5-*O*-di-glucoside (22.4%), followed by degradation of cyanidin-3,5-*O*-di-glucoside (18.3%), delphinidin-3-*O*-glycoside (12.5%), and cyanidin-3-*O*-glucoside (6.65%) (Figure 1a).

The same trend could be observed for the degradation of a mixture of anthocyanin standards after heating the mixture for 20 min. However, the degradation of individual anthocyanins was lower than in pomegranate juice (Figure 1b), indicating an interaction effect of the compounds in the pomegranate juice. At half of the thermal treatment period (10 min, 80 °C), the content of diglycosylated anthocyanidins was the same (about 10%) and considerably lower for delphinidin-3-*O*-glycoside and cyanidin-3-*O*-glucoside (below 10%).

In the mixture of anthocyanin standards, then the degradation of the same compounds in pomegranate juice, except cyanidin-3-*O*-glucoside, reached similar levels at 10 and 20 min of treatment. Furthermore, amongst the observed compounds, the higher levels of degradation of diglycosylated anthocyanins were reflected in all environments and at all treatment times.

Thermal treatment of pomegranate juice on a bigger scale, on a homeware induction cooking plate in an open stainless still pot, revealed that after initial 10 min heating at 80 °C, degradation of delphinidin-3,5-O-diglucoside was lowest and highest for cyanidin-3-*O*-glucoside, 17.7% and 32.7%, respectively (Figure 2). After 3 and 5-month storage of thermally treated pomegranate juice, further degradation was observed for all analyzed anthocyanins. Delphinidin-3,5-O-diglucoside degradation was the lowest of all and ranged from 17.7% after initial 10 min heating at 80 °C to 49.1% and 67.2% after 3 and 5 months of storage of thermally treated pomegranate juice. In contrast to what we noticed in the short term (Figure 1), the highest and almost the same degradation were observed for monoglycosylated anthocyanidins (Figure 2).

The degradation of delphinidin-3-*O*-glucoside and cyanidin-3-*O*-glucoside did not change significantly after 3 or 5 months of storage. After 5 months of storage of thermally treated pomegranate juice, almost all delphinidin-3-*O*-glucosides and cyanidin-3-*O*-glucosides were degraded, 91.7% and 92.3%, respectively.

Our findings are in agreement with other studies, except that we followed the degradation rate of individual anthocyanin in pomegranate juice. It was found that the anthocyanin loss was 15.4% to 28.3% after 2 min thermal treatment of pomegranate juice at 90 °C [44] but after conventional thermal processing with mild temperature-long time treatment at 65 °C for 1 min it was found that anthocyanin content was enhanced [45].

### 3.6. Color Measurements

Changes in color parameters were monitored during storage. These color measurements were performed immediately after thermal treatment and 3 or 5 months of storage. Data are shown in Table 8.

The color measurements for the initial thermally treated juice are in accordance with the values reported for freshly squeezed (untreated) as well as hydrostatically treated pomegranate juices [46,47]. Furthermore, after 3 to 5 months in storage, the juice underwent a clearly apparent color change, indicated by the change in color parameters (change in color is defined at ΔE > 5). Further storage from 3 to 5 months did not significantly affect the color and the color intensity remained practically unchanged (ΔE_3,5_ = 1.0, ΔC_3,5_ = 0.2). The initial change in color can be attributed to clarification. Indeed, the final brightness and chromatic attributes of our pomegranate juice samples were very similar to the reported color properties that were achieved by the addition of clarification agents to fresh pomegranate juice, followed by subsequent microfiltration or ultrafiltration [48].

### 3.7. Fibrous Compounds in Pomegranate Peel

Pomegranate peel is the predominant biomass fraction of pomegranate fruit rich in fibers. Thus, we further examined this fraction to learn more about the nature of these fibers and possible pathways of subsequent valorization. Furthermore, the initial plant production residues were characterized for their fibrous and bioactive content, and the amount of material that might be removed during the extraction step of bioactive was determined (Table 9). In softwood, which is commonly used in papermaking, the cellulose content can exceed 40%, hemicellulose content can exceed 30% and lignin content is around 20–35%, therefore having a 1.25:1:1 cellulose/hemicellulose/lignin ratio [49].

In our analysis, the pomegranate peel exhibited up to 75% lower content of these compounds, whereas the cellulose/hemicellulose/lignin ratio was maintained roughly at 1:1:1. This ratio and the polymer content are relatively similar to the values reported for citrus or banana peels [50,51]. Compared to wood, pomegranate peel contained more than ten times higher amounts of extractives, as two-thirds of the peel’s dry matter could be removed by the extraction procedure. Nevertheless, this plant residue could benefit from the extraction in aqueous and/or ethanol solvents [36], perhaps being potentially interesting for cascading where in the first step bioactives are extracted and in the second step, the fibers in the remaining biomass are valorized.

The mechanical and optical characteristics were measured upon isolation of the fibers. Pomegranate peels were comprised of wide fibers with pronounced fibrillation. Overall, the fibers were short compared to deciduous and coniferous trees and comparable or even longer to various non-wood fibers such as stems, stalks and straw of cultivated plants. It is well known that curled fibers produce papers with poorer tensile properties while increasing resistance to fracture and tear. Compared to softwood, the pomegranate peel fibers present a 50% to 60% lower curl, being more in line with hardwood or alternative non-wood pulps [52,53,54].

### 3.8. Test Production of Paper

Paper sheets were prepared to simulate the industrial process of paper production (Table 10). Due to the initially low cellulose content c and a high percentage of extractives, the paper made from 100% pomegranate peel exhibited charring and fracturing and was unsuitable for further characterization. To enable the potential incorporation of these processing residues into papermaking, substituting a part of wood-derived commercially available cellulose with the pomegranate peel was then studied. As expected, the addition of pomegranate to cellulose impaired mechanical properties (tension, tensile strength, and cracking strength). Furthermore, some new visual features were produced, such as reduced whiteness, increased opacity and roughness, and an interesting texture (Appendix A).

The comparison of the characteristics of paper produced from the combination of pomegranate peels and cellulose with the optimal characteristics of other basic paper and cardboard products shows that incorporating the examined residues can introduce multifunctional features to pure cellulose. The final mixed product was best suited for a variety of paper accessories (office paper, and particularly for envelopes and gift papers where the visual properties could be utilized), as well as eventual paper for newspapers and magazines or marketing leaflets/insets.

Furthermore, we can also conclude that pomegranate peels need to be combined with cellulose as raw materials for papermaking and are also less suitable for producing cardboard, mainly due to low grammage.

## 4. Conclusions

This contribution demonstrates the polyphenol and antioxidative profiles in different fruit parts of an unknown cultivar of pomegranate grown in the northern part of the Mediterranean, Northwestern Istria, as well as further pathways of valorization of processing residues, such as the recovery of bioactive components and utilization of the remaining fibrous compounds in, e.g., paper packaging. The results have confirmed that there are significant differences reflected in the content of anthocyanins in comparison to other worldwide grown pomegranate cultivars. The results will be useful for future studies to understand the similarities and differences in biochemical properties of different pomegranate cultivars as well as the recovery and utilization of bioactive compounds from pomegranate agroindustrial wastes.

## Figures and Tables

**Figure 1 foods-11-02740-f001:**
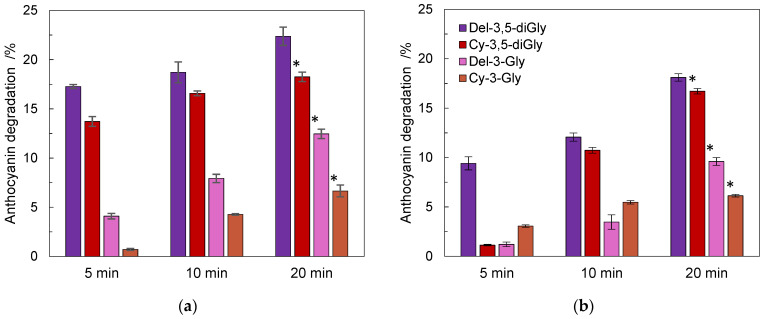
Effect of the duration of thermal treatment at 80 °C on the anthocyanin (delphinidin-3,5-*O*-di-glucoside, cyanidin-3,5-*O*-di-glucoside, delphinidin-3-*O*-glucoside, and cyanidin-3-*O*-glucoside) degradation in pomegranate juice (**a**) and in the mixture of anthocyanin standards (**b**). Data are means ± SD. *, indicates dissimilarity versus 5–20 min standardized degradation rate of delphinidin-3,5-*O*-di-glucoside (f2 < 50).

**Figure 2 foods-11-02740-f002:**
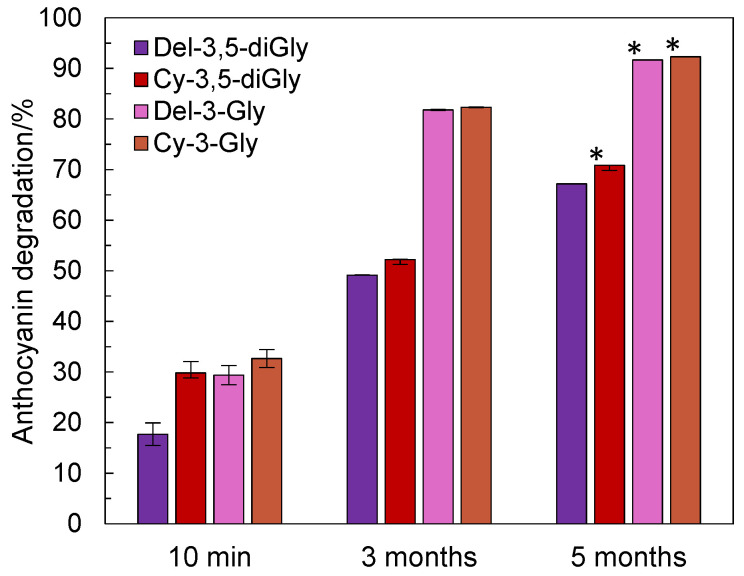
Degradation of anthocyanins (delphinidin-3,5-*O*-di-glucoside, cyanidin-3,5-*O*-di-glucoside, delphinidin-3-*O*-glucoside and cyanidin-3-*O*-glucoside) in pomegranate juice after thermal treatment (10 min 80 °C) and after long-term storage for either 3 or 5 months at 18 °C in sealed containers in the dark. Data are means ± SD. *, indicates dissimilarity versus 10 min—5 months degradation profile of delphinidin-3,5-*O*-di-glucoside (f2 < 50).

**Table 1 foods-11-02740-t001:** Correlation factor, the limit of detection (LOD), and limit of quantification (LOQ) of anthocyanins standards by HPLC method.

Compound	Regression Equation	Correlation Factor, R^2^	LOD (µg/mL)	LOQ (µg/mL)
Cy-3-Gly	y = 72.79·x − 42.09	0.9999	0.585	1.77
Cy-3,5-diGly	y = 79.05·x − 24.20	0.9997	1.38	4.16
Del-3-Gly	y = 138.2·x − 116.2	0.9997	0.90	2.73
Del-3,5-diGly	y = 77.38·x − 25.05	0.9997	1.35	4.09
Pel-3-Gly	y = 122.9·x − 126.7	0.9979	2.52	7.62
Pel-3,5-diGly	y = 35.10·x − 9.443	0.9984	1.95	5.93

y = peak area; x = concentration in µg/mL.

**Table 2 foods-11-02740-t002:** Fresh mass, water content, and obtained lyophilized water and 70% ethanol extracts of the pomegranate peel, mesocarp, and arils (before and after juicing).

Source	Fresh Mass Fraction	Water Content	Lyophilized Water Extract	Lyophilized Ethanol Extract
	% (FM/FM)		% (DM/FM)	
Peel	18.9	72.6	43.2	62.8
Mesocarp	30.8	73.9	64.9	75.4
Arils	50.2	n.d.	n.d.	n.d.
- seeds and pulp	18.1	n.d.	80.5	82.3
- juice *	32.1	80.1	20.6	28.6

FM—fresh matter of the samples, DM—dry matter of the lyophilized extract, n.d.—not determined, * ρ = 1.056 g/mL.

**Table 3 foods-11-02740-t003:** Polyphenol group profile and antioxidant capacity of water and 70% ethanol extracts of pomegranate fruit fractions.

Source	Water Extracts				Ethanol Extracts			
	Total Phenolic Content (mg GAE/g DM)	Flavonoids (mg CE/g DM)	Antioxidant Capacity (µg TE/g DM)	Flavonoids: Phenolics Ratio (/)	Total Phenolic Content (mg GAE/g DM)	Flavonoids (mg CE/g DM)	Antioxidant Capacity (µg TE/g DM)	Flavonoids: Phenolics Ratio (/)
Peel	8.80 ± 0.40 ^a1^	2.80 ± 0.20 ^a2^	128 ± 2 ^a3^	0.32	30.5 ± 0.6 ^a4^	4.25 ± 0.05 ^a5^	17.0 ± 6.0 ^a6^	0.14
Mesocarp	8.00 ± 0.20 ^a1^	1.98 ± 0.01 ^a2^	69.7 ± 0.7 ^b3^	0.25	26.3 ± 0.0 ^b4^	2.50 ± 0.04 ^b5^	5.50 ± 0.20 ^ab6^	0.095
Juice	1.74 ± 0.02 ^b1^	0.083 ± 0.009 ^b2^	23.0 ± 1.0 ^c3^	0.048	1.12 ± 0.04 ^c4^	0.100 ± 0.006 ^c5^	2.75 ± 0.03 ^ac6^	0.084
Seeds	0.420 ± 0.010 ^c1^	0.031 ± 0.001 ^b2^	0.266 ± 0.003 ^d3^	0.074	1.48 ± 0.02 ^d4^	0.220 ± 0.020 ^c5^	4.90 ± 0.20 ^ab6^	0.015

GAE—gallic acid equivalent, CE—catechin equivalent, TE—Trolox equivalent, DM—dry matter of the lyophilized extract. Data are means ± SD. Superscript numbered letters indicate significant differences (*p* < 0.01) between means in individual columns.

**Table 4 foods-11-02740-t004:** Polyphenol contents in various reported pomegranate extracts.

Origin	Pomegranate Part	Pretreatment and Extraction Conditions	Total Phenolic Content	Source
Peru, unknown cultivar	Peel	Pre-steamed, 80% MeOH, 0.1% HCl	101.86 ± 12.8 mg GAE/g DM extract	[19]
	Mesocarp		198.17 ± 2.9 mg GAE/g DM extract	
	Juice 1	Juicing at 10 bar	2015.2 ± 21.66 mg GAE/L	
	Juice 2	Juicing at 150 bar	5186.0 ± 172.5 mg GAE/L	
	Juice 3	Juicing at 250 bar	2122.0 ± 0.0 mg GAE/L	
Tunisia, 12 cultivars	Peel + mesocarp	Pre-homogenized, 80% ETOH + 70% ACE	205.07 ± 0.0 to 276.35 ± 0.07 mg GAE/g extracted	[37]
Iran, local markets	Peel − undefined	sonicated in 70% EtOH, 60 °C, 30 min	86.78 mg GAE/g DM extract	[38]
		70% EtOH, 30 °C, 30 min	76.22 mg GAE/g DM extract	
		70% EtOH, 30 °C, 10 min	70.95 mg GAE/g DM extract	
		30% EtOH, 30 °C, 10 min	49.35 mg GAE/g DM extract	
Turkey, 4 cultivars	Peel − undefined	Pre-homogenized, 50% EtOH	1.78 to 3.55 mg GAE/g fresh weight	[39]
	Seeds		1.31 to 1.55 mg GAE/g fresh weight	
	Juice		0.121 to 0.177 mg GAE/g fresh weight	
Iran, 9 cultivars	Peel − undefined	Soxhlet extraction in ACE, EtOAc, MeOH, and H_2_0	18.61 ± 0.53 to 36.40 ± 1.34 mg GAE/g extract	[40]
	Pulp − undefined		11.62 ± 0.63 to 21.03 ± 1.51 mg GAE/g extract	

MeOH—methanol, EtOH—ethanol, ACE- acetone, EtOAc—ethyl acetate, GAE—gallic acid equivalent, DM—dry matter of the lyophilized extract. Data are shown as available.

**Table 5 foods-11-02740-t005:** The concentration of catechin and the presence (+) of phenolic compounds in the dry extracts prepared with 70% ethanol.

Source	Catechin (mg/g DM)	Quercetin (R_F_ = 0.43)	*o*-Coumaric Acid (R_F_ = 0.61) *	Gallic Acid (R_F_ = 0.27) *	Caffeic Acid (R_F_ = 0.43) *	Chlorogenic Acid (R_F_ = 0.41) *
Peel	0.429	+	+	+	+	+
Mesocarp	0.083	+	+	+	+	+
Juice	0.024	+	+	+	+	+
Seeds	0.088	+	+	+	+	+

*—detected only after 18-h hydrolysis of pomegranate extracts, R_F_—retardation factor, DM—dry matter of the lyophilized extract.

**Table 6 foods-11-02740-t006:** Anthocyanin profiles of water and ethanol pomegranate extracts.

Source Peak Number	Retention Time (min)	Molecular Ion [M]^+^ (m/z)	MS/MS Fragment Ions (m/z)	Identification
Peel				
1	7.98	611	449, 287	Cyanidin + 2 hexoses
2	8.62	465	303	Delphinidin 3-glucoside
3	9.27	595	433, 271	Pelargonidin + 2 hexoses
4	9.96	449	287	Cyanidin 3-glucoside
5	11.37	433	271	Pelargonidin 3-glucoside
6	12.65	419	287	Cyanidin + arabinose or Cyanidin + xylose
Mesocarp				
1	9.97	449	287	Cyanidin 3-glucoside
2	11.41	433	271	Pelargonidin 3-glucoside
Juice				
1	6.66	627	465, 303	Delphinidin + 2 hexoses
2	8.00	611	449, 287	Cyanidin + 2 hexoses
3	8.65	465	303	Delphinidin 3-glucoside
4	9.28	595	433, 271	Pelargonidin + 2 hexoses
5	10.00	449	287	Cyanidin 3-glucoside
6	11.40	433	271	Pelargonidin 3-glucoside
7	12.68	419	287	Cyanidin + arabinose or Cyanidin + xylose

**Table 7 foods-11-02740-t007:** Individual anthocyanins content in peel and juice pomegranate extracts determined with HPLC.

Anthocyanin	Water Extracts		Ethanol Extracts	
	Peel (µg/g DM)	Juice (µg/g DM)	Peel (µg/g DM)	Juice (µg/g DM)
Del-3,5-diGly	n.d.	22.07 ± 0.09 ^a1^	n.d.	16.2 ± 0.1 ^b1^
Cy-3,5-diGly	378.3 ± 0.1 ^a2^	31.1 ± 0.1 ^b2^	23.45 ± 0.02 ^b2^	28.9 ± 0.3 ^b2^
Del-3-Gly	9.03 ± 0.01 ^a3^	7.10 ± 0.04 ^a3^	9.83 ± 0.04 ^a3^	7.49 ± 0.05 ^a3^
Pel-3,5-diGly	270.88 ± 0.02 ^a4^	1.35 ± 0.03 ^b4^	23.51 ± 0.03 ^a4^	0.80 ± 0.03 ^b4^
Cy-3-Gly	1029.0 ± 0.3 ^a5^	21.52 ± 0.03 ^bc5^	127.13 ± 0.08 ^b5^	23.0 ± 0.1 ^bc5^
Pel-3-Gly	490.27 ± 0.03 ^a6^	1.08 ± 0.01 ^bc6^	58.84 ± 0.01 ^b6^	1.03 ± 0.05 ^bc6^
Sum	2177.5 ± 0.3	84.3 ± 0.1	242.76 ± 0.01	77.4 ± 0.3

n.d.—not detected. DM—dry matter of the lyophilized extract. Superscript numbered letters indicate significant differences (*p* < 0.01) between means in individual rows.

**Table 8 foods-11-02740-t008:** Color parameters (L*, a*, and b*), cumulative difference in color (ΔE) and color intensity (ΔC) of pomegranate juice that was treated for 10 min at 80 °C and stored at 18 °C.

Parameter	Initial	3 Months Storage	5 Months Storage
L*	26.1 ± 0.7 ^a1^	19.1 ± 0.6 ^b1^	18.1 ± 0.7 ^b1^
a*	13.0 ± 0.5 ^a2^	1.4 ± 0.1 ^b2^	1.2 ± 0.2 ^b2^
b*	3.7 ± 0.2 ^a3^	1.0 ± 0.2 ^b3^	0.8 ± 0.1 ^b3^
ΔE	0	13.9	14.6
ΔC	0	11.9	12.2

L*—CIE brightness; CIE chromaticity: a* CIE red(+)/green(−), b* CIE yellow(+)/blue(−). Superscript numbered letters indicate significant differences (*p* < 0.01) between means in individual rows.

**Table 9 foods-11-02740-t009:** Composition of natural fibers in pomegranate peels and mechanical and optical characteristics of the isolated fibers.

Parameter	Pomegranate Peels
Cellulose (% DM)	11.0
Hemicellulose (% DM)	11.6
Lignin (% DM)	12.2
Extract in 70% ethanol (% DM)	68.2
Fiber width (µm)	39.82
Curl (%)	9.91
Fibrillation (%)	4.45
Lc(l) ISO (µm)	112

DM—a dry matter of the peel. Fiber width curl and fiber length (Lc(Lc(l) ISO) are given as length-weighted values. Fibrillation is given as a percentage of the projection area of fibrils to the projection area of the entire object.

**Table 10 foods-11-02740-t010:** Optical and mechanical properties of paper.

Parameter	Samples	
	Pomegranate Peels/Cellulose (15/85%)	Cellulose (100%)
Grammage (g/m^2^)	63.6	65.0
Thickness (um)	161	116
Tensile index (Nm/g)	41.4	53.3
Breaking length (km)	4.222	5.334
Bendtsen roughness (mL/min)	1677	342
ISO whiteness (%)	41.4	77.0
Opacity (%)	96.8	86.6
Tear index (mNm^2^/g)	7.35	7.85
Burst index (KNm/g)	2.65	3.48

Cellulose composition (80% eucalyptus, 20% conifers, ground to 30 SR).

## Data Availability

The data presented in this study are available on request from the corresponding author.

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
