# Peer review of "Extraction of Polyphenols and Valorization of Fibers from Istrian-Grown Pomegranate (Punica granatum L.)"

_foods, 2022, doi:10.3390/foods11182740_

Round 1

Reviewer 1 Report

Dear editor,

The work appears to be comprehensive; the introductory section provides sufficient background; however, some references have more than 10 years and should be updated.The materials and methods section is adequate and allows the authors to achieve their results. Nevertheless, the anthocyanin determination was performed by a previous methodology using wheat grains as matrix; therefore, since the matrix is different, a validation procedure must be performed to determine the anthocyanin content accurately. The results are clear and the conclusion is based on the findings. However, the format iof the tables must be correct as table 6 and the number of samples should be added in the tables and figures.

Minor comments:

-Section 2.3: Were the freeze-dried fruits ground before sampling?

-Add reference in section 2.16.

Author Response

We would like to thank the reviewer for detailed reviewing our manuscript.

Rev 1 comment: “the work appears to be comprehensive; the introductory section provides sufficient background; however, some references have more than 10 years and should be updated”.

Response: We updated references in the Introduction section. For the clarification of the purpose of the research, we also added the lines “In the past few years, the awareness of health beneficial effects of pomegranate fruit has risen among consumers, which consequently led to higher demand for pomegranate fresh fruit as also pomegranate-based processed products. From 2013 to 2017 total import volume to the EU increased by 28,000 tonnes( https://unece.org/sustainable-development/press/new-unece-standard-will-boost-international-trade-pomegranate). Regarding the high demand for pomegranate fruit, some countries even plan to increase the total area of pomegranate orchards [21]”

Changes could be seen in “Track change” mode.

 Rev 1 comment: “Nevertheless, the anthocyanin determination was performed by a previous methodology using wheat grains as matrix; therefore, since the matrix is different, a validation procedure must be performed to determine the anthocyanin content accurately”.

Response: LC-MS described in section 2.11 was used only for qualitative analysis of anthocyanins.  Anthocyanins content was determined with the HPLC method, described in section 2.12. We used external standards cyanidin-3-O-glucoside (Cy-3-Gly), cyanidin-3,5-di-O-glucoside (Cy-3,5-diGly), pelargonidin-3-O-glucoside (Pel-3-Gly), pelargonidin-3,5-di-O-glucoside (Pel-3,5-diGly), delphinidin-3-O-glucoside (Del-3-Gly), delphinium-3,5-di-O-glucoside (Del-3,5-diGly) in the concentration range from 0.5 µg/mL to 200 µg/mL. For each standard, the 5-point calibration curve was prepared in the same matrix as pomegranate extracts, and each solution for the 5-point calibration curve was analysed in triplicate. The linear regression and correlation factor R2 were determined in Excel 2017 with the Data Analysis tool, ANOVA programme. LOD and LOQ were calculated according to the International Council for Harmonisation of Technical Requirements for Pharmaceuticals for Human Use (ICH) guidance on the validation of analytical procedures. LOD was expressed as 3.3xs/S and LOQ as 10xs/S, where s is the standard deviation of response and S is the slope of the calibration curve. In the table below, R2, LOD and LOQ are represented. These data were not included in the manuscript.

Table: Linear correlation factor, R2, LOD and LOQ for anthocyanins standards used for HPLC determination in section 2.12.

Standard

Linear regression correlation factor, R2

LOD (µg/mL)

LOQ

(µg/mL)

Cy-3-Gly

0,9999

0.585

1.77

Cy-3,5-diGly

0.9997

1.38

4.16

Del-3-Gly

0.9997

0.90

2.73

Del-3,5-diGly

0.9997

1.35

4.09

Pel-3-Gly

0.9979

2.52

7.62

Pel-3,5-diGly

0.9984

1.95

5.93

Rev 1 comment: “the format iof the tables must be correct as table 6 and the number of samples should be added in the tables and figures.”.

Response: In the title of Table 6 “Individual anthocyanins content in peel and juice

 pomegranate extracts determined with HPLC, shown in the order of peak assignment”,  text  “ shown in the order of peak assignment” was deleted in line 478 and peek assignments are named in Figure S1.

Rev 1 comment: “Section 2.3: Were the freeze-dried fruits ground before sampling?”

Response: In section 2.3 of the manuscript the preparation of the lyophilized extracts is described in lines 123 to 125 and it is written “For the peels, mesocarp and seeds, the lyophilized material was crushed (A 11 analytical mill, IKA, Staufen im Breisgau, Germany) to a fine powder prior to extraction, whereas the lyophilized juice was extracted without crushing.”

Rev 1 comment: “Add reference in section 2.16.”

Response: In the section 2.16 reference [35]: DOI: 10.1007/s00107-021-01779-y was added.

Reviewer 2 Report

Dear authors.

The introduction does not provide enough information about the scientific data related to the aim of this article.

Statistical analysis of the results is not sufficient.

The results section presents a summary of other authors' data in a table, which confuses the reader.

In 403-412 lines presented information referred to the methodology.

From Table 4 data is not clear what extracts were analysed.

In some article parts, the discussion is very poor.

Author Response

We would like to thank the reviewer 2 for comprehensive review of our manuscript.

Rev 2 comment: “The introduction does not provide enough information about the scientific data related to the aim of this article”.

            Response: In the introduction, references were updated according to the scientific data related to the aim of this article.

Rev 2 comment: “Statistical analysis of the results is not sufficient.”

            Response: All results values represent the mean ± standard deviation of three sampling replicates. For the anthocyanins determination, the linear regression and correlation factor R2 were determined in Excel 2017 with the Data Analysis tool, ANOVA programme. LOD and LOQ were calculated according to the International Council for Harmonisation of Technical Requirements for Pharmaceuticals for Human Use (ICH) guidance on the validation of analytical procedures. LOD was expressed as 3.3xs/S and LOQ as 10xs/S, where s is the standard deviation of response and S is the slope of the calibration curve.

Rev 2 comment: “The results section presents a summary of other authors' data in a table, which confuses the reader.”

Response: In line 322 title of the Section 3 was changed from Results to “Results and Discussion”. Table 3 was meant as a part of the discussion to be able to compare our findings with polyphenol content in various reported pomegranate extracts.

Rev 2 comment:In 403-412 lines presented information referred to the methodology.”

Response: In the manuscript, in lines 403 to 407 it is written “In another assay system, Gil et al. [39] also showed different TEAC values for different groups of phenolic compounds in pomegranate juice. The TEAC value decreased in the order from punicalagins > anthocyanins > hydrolyzable tannins > ellagic acids. In the present study, we can further conclude from tables 2 and 6 that TEAC values correlate with flavonoid content, more specifically with anthocyanins, rather than total phenolic content.” Whit this part we compared our correlation of flavonoid content with the reported one and we did not describe any methodology.

Lines, 408 to 413, referred to as lines that present information about the methodology were deleted in the manuscript. Lines, 417 to 419, also referred to as lines that present information about the methodology were deleted in the manuscript.

 Rev 2 comment:From Table 4 data is not clear what extracts were analysed.”

Response: In lines, 426 to 427 is written “Table 4. The concentration of catechin and the presence (+) of phenolic compounds in the dry extracts prepared with 70% ethanol”.

Rev 2 comment:In some article parts, the discussion is very poor.”

Response: We have expanded the discussion with additional explanations in section 3.2.

Reviewer 3 Report

The manuscript entitled "Extraction of Polyphenols and Valorization of Fibers from Istrian-grown Pomegranate (Punica granatum L.)" presents the characterization of different parts of pomegranate fruit. Besides there is no great novelty in these results (phenolic composition and antioxidant capacity), there is also no explanation why the authors have characterized the anthocyanidin content by LC-MS and HPLC and other classes of phenolic compounds by TLC. The only novelty of the study is the application of pomegranate wastes for paper production. At least, a complete characterization of the extracts produced should be carried out by LC-MS.

Author Response

We would like to thank the reviewer for many good suggestions.

Rev 3 comment:there is no great novelty in these results (phenolic composition and antioxidant capacity)”

Response: Consumers' awareness of pomegranate fruit as a “super-food”, led to higher demand for pomegranate fruit and other pomegranate products. As a result, worldwide production of pomegranate fruit is increasing. Through the review of  MDPI publications about pomegranate, we found out that MDPI publications encourage research about new pomegranate cultivars, so we were encouraged to send our manuscript to Foods. The manuscript “Extraction of Polyphenols and Valorization of Fibers from Istrian-grown Pomegranate (Punica granatum L.)” is focused on the new pomegranate cultivar, grown in the most northern part of the Medeterranean, Northwestern Istria. To our knowledge, there is little known about pomegranate grown in the Northwestern Istria, although there are pomegranate plantations, some new ones were established for the purpose to increase pomegranate production in this region. For this region, the knowledge about the new pomegranate cultivar is valuable as well as the novelty of shifting the conventional linear pomegranate production schemes to a circular economy, an approach which can improve overall efficiency and resource use, as well as provide new functional added-value products. Comparison of anthocyanins' thermostability and degradation during the long period in standard mixtures and real pomegranate juice to our knowledge also contribute to the novelty of the overall knowledge about pomegranate as well to the juice-production sustainability.

Rev 3 comment:there is also no explanation why the authors have characterized the anthocyanidin content by LC-MS and HPLC and other classes of phenolic compounds by TLC. At least, a complete characterization of the extracts produced should be carried out by LC-MS.”

Response: Main products of the pomegranate presented in the manuscript were pomegranate juice and a new functional added-value product, paper from pomegranate peel. Anthocyanins are natural pigments, phenol compounds, which contribute to the beautiful colour of pomegranate juice as well as influence the colour of the paper from pomegranate peel. Because of this reason, we used LC-MS for the qualitative confirmation of anthocyanins as described in section 2.7. and HPLC for the quantitative determination of anthocyanins. Because other classes of phenolic compounds do not contribute to the colour but can influence the antioxidative activity of juice as well to the new functional added-value product, paper from pomegranate peel,  they were first qualitatively analysed with TLC.   We are aware that LC-MS of all phenolic compounds could improve our research, but due to the lack of the exact samples after we finished with the research, unfortunately, we were not able to even start with LC-MS.  And as TLC is still used for qualitative determination of phenolic compounds from pomegranate and was recently published in Foods (https://doi.org/10.3390/foods11081070), we hope that our research will satisfy the criteria for publishing in Foods.

Round 2

Reviewer 1 Report

Dear editor,

The authors have adressed all issues and I believe the MS is suitable for publication.

Best regards,

Reviewer

Author Response

Thank you for reviewing our manuscript.

Reviewer 2 Report

Dear authors.

Used statistical analysis methods have to be described in the separate abstract in the manuscript.

Author Response

Rev 2 comment: Used statistical analysis methods have to be described in the separate abstract in the manuscript.

Response: We thank the reviewer for the opportunity to make the manuscript more complete. According to Rev2 suggestion, lines 262 to 270 with Table 1 were added with the description of calibration curves for individual anthocyanins:

For each standard, the 5-point calibration curve was prepared in the same matrix as pomegranate extracts, and each solution for the 5-point calibration curve was analysed in triplicate. For each anthocyanin standard the linear regression correlation factor, R2, was determined as well as the limit of detection, LOD, and limit of quantification, LOQ (Table 1).

Table 1. Correlation factor, the limit of detection (LOD) and limit of quantification (LOQ) of anthocyanins standards by HPLC method.

Compound

Regression equation

Correlation factor, R2

LOD (µg/mL)

LOQ

(µg/mL)

Cy-3-Gly

  y = 72.79·x - 42.09

0,9999

0.585

1.77

Cy-3,5-diGly

y = 79.05·x - 24.20

0.9997

1.38

4.16

Del-3-Gly

y = 138.2·x - 116.2

0.9997

0.90

2.73

Del-3,5-diGly

y = 77.38·x - 25.05

0.9997

1.35

4.09

Pel-3-Gly

y = 122.9·x - 126.7

0.9979

2.52

7.62

Pel-3,5-diGly

y = 35.10·x - 9.443

0.9984

1.95

5.93

                 y = peak area; x = concentration in µg/mL

 Statistical analysis methods were described in a separate added section 2.17 and results in Tables 3, 7 and 8 were updated with statistical parameters. Figures: Figure1a, Figure 1b and Figure2 were replaced with results, updated with statistical parameters.

Reviewer 3 Report

Although I understand the authors' reasons for not having performed HPLC-MS analysis of the other classes of phenolics, I do not think TLC analysis is acceptable. Can't the authors prepare new extracts and analyse them by HPLC-MS?

Author Response

Responses to the reviewer's  Rev 3 comments – Round 2

Rev 3 comment: Although I understand the authors' reasons for not having performed HPLC-MS analysis of the other classes of phenolics, I do not think TLC analysis is acceptable. Can't the authors prepare new extracts and analyse them by HPLC-MS?

Response to Rev3 comment:

We are aware that each analytical technique has advantages and disadvantages. High-performance thin-layer chromatography (HPTLC) has several advantages compared to HPLC. We were using one of these advantages, which enabled us to perform post-chromatographjic derivatization on the HPTLC plate under equal conditions for all the extracts and standards. Special instrumental equipment - rearly available in the labs - is needed to perform post-column derivatization in HPLC. The HPTLC- densitometric method, which included post-chromatographic derivatization with DMACA detection reagent (specific for flavan-3-ols and proanthocyanidins [26]) is very sensitive and enabled quantification of catechin in the range from 2 to 30 ng on the plate. LOD for catechin is 0.2 ng on the plate. In section 2.7 lines from 187 to 188 were rephrased and lines from 191 to 192 were added with the description of the catechin calibration curve used for the determination of catechin in pomegranate ethanol extracts, analysed with HPTLC and HPTLC densitograms of standard catechin and ethanol pomegranate extracts was added as supplementary Figure S1 (attached). As a result of these additional results being shown we have also added an additional coauthor to the manuscript.

HPTLC method for screening of phenolic compounds (flavonoids and phenolic acids) enabled monitoring of all the samples under equal conditions before and after derivatization with NST reagent, which enabled fast monitoring of all the samples under the same conditions.

Further, in section 3.2 lines from 443 to 447 we added text “Qualitative and quantitative analyses of flavan-3-ol catechin in 70% ethanol(aq) pomegranate extracts were performed on cellulose stationary phase by HPTLC-densitometric method, which included post-chromatographic derivatization with DMACA detection reagent (specific for flavan-3-ols and proanthocyanidins [26]). Catechin was detected in all extracts (see Figure S1 attached)” and changes were made in lines from 447 to 448,  from 452 to 453 and from 457 to 459.

All HPTLC methods applied in our study were applied also for studies of other plant extracts as it is evident from the literature.
